

# The double high tide at Port Ellen: Doodson's criterion revisited

Hannah A. M. Byrne[1], J. A. Mattias Green[1], and David G. Bowers[1]

[1]Bangor University, School of Ocean Sciences, Menai Bridge, Anglesey, LL59 5AB, UK

*Correspondence to:* Dr Mattias Green (m.green@bangor.ac.uk)

**Abstract.** Doodson proposed a criterion to predict the occurrence of double high (or double low) waters when a higher frequency tidal harmonic is added to the semi-diurnal tide. If the phasing of the harmonic is optimal, the condition for a double high water can be written $bn^2/a > 1$ where $b$ is the amplitude of the higher harmonic, $a$ is the amplitude of the semi-diurnal tide and $n$ is the ratio of their frequencies. Here we expand this criterion to allow for (i) a phase difference $\phi$ between the semi-diurnal tide and the harmonic and (ii) the fact that the double high water will disappear in the event that $b/a$ becomes large enough for the higher harmonic to be the dominant component of the tide. This can happen, for example, at places or times where the semi-diurnal tide is very small. The revised parameter is $br^2/a$, where $r$ is a number generally less than $n$, although equal to $n$ when $\phi = 0$. The theory predicts that a double high tide will form when this parameter exceeds 1 and then disappear when it exceeds a value of order $n^2$ and the higher harmonic becomes dominant. We test these predictions against observations at Port Ellen in the Inner Hebrides of Scotland. For most of the data set, the largest harmonic of the semi-diurnal tide is the 6th diurnal component, for which $n = 3$. The principal lunar and solar semi-diurnal tides are about equal at Port Ellen and so the semi-diurnal tide becomes very small twice a month at neap tides. A double high water forms when $br^2/a$ first exceeds a minimum value of about 1.5 as neap tides are approached and then disappears as $br^2/a$ then exceeds a second limiting value of about 10 at neap tides in agreement with the revised criterion.

## 1 Introduction

Double high waters in the semi-diurnal tide, seen at a select few locations globally, are an intriguing and sometimes puzzling feature of coastal tidal observations. In a double high water, the tide rises to a first maximum followed by a short dip in water level before it rises again to a second maximum and then falls towards the subsequent low water. The most famous and best studied example of a double high water is that at Southampton on the south coast of England, where the extended period of deep water associated with the double high tide gave the port a commercial advantage over its rivals. Other examples of double high waters are found at Den Helder in the Netherlands and Woods Hole, USA. An analogous phenomenon, a double low water, is seen at Providence in Narragansett Bay, USA (Redfield, 1980).

No explanation of double high (or low) waters can be found in the direct action on the ocean of the tidal forces of the moon and sun. To create a double high water in the semi-diurnal tide it is necessary to add a higher frequency oscillation with the correct phase and sufficient amplitude. Higher harmonics of the semi-diurnal tide (that is oscillations with a frequency higher than twice per day) are created as the tidal wave enters shallow water, either through the reduction of water depth or through





the enhanced effects of quadratic bottom friction (Godin, 1988). Transient seiches created by the tide can also serve to provide the right conditions for double tides (Bowers et al., 2013).

The amplitude and phase of the higher frequency oscillation must meet certain conditions to produce a double high water. The simplest of these conditions was described by Doodson and Warburg (1941). If the frequency of the oscillation is $n$ times

that of the semi-diurnal tide and the oscillation has a trough exactly coinciding with the peak of high water, a double high water will form if

$$\frac{bn^2}{a} > 1 \tag{1}$$

where $b$ is the amplitude of the oscillation and $a$ the amplitude of the semi-diurnal tide. For example, if the oscillation has the period of a quarter of a day, then $n = 2$ and the amplitude must be at least 1/4 that of the semi-diurnal tide to produce a double

high water. This is a stringent requirement: in most cases, the quarter diurnal component will not be big enough on its own to produce a double high water and additional higher harmonics will be required. As the frequency, $n$, of the higher harmonic increases, inequality (1) shows that the required amplitude ratio, $b/a$, becomes smaller.

It is worth noting here that the amplitude, $a$, of the semi-diurnal tide in (1) is that observed on the day, i.e., it is the sum of the lunar and solar semi-diurnal tidal constituents on that day. The value of therefore changes through the fortnightly springs-neaps

cycle. The semi-diurnal tide on a particular day, changing in amplitude from day to day in this way is called the D2 tide (Pugh, 1987). Similarly, the amplitude of the higher frequency oscillation, $b$, is also that observed on the day and will be different on other days. If the higher frequency oscillation is a harmonic of the semi-diurnal tide it is referred to as a D4 tide in the case of a quarter diurnal oscillation, D6 if the period is 4 hours and so on. One way to increase the value of the ratio $b/a$ in the Doodson criterion is to reduce the amplitude, $a$, of the D2 tide. A good place to look for double high waters, therefore, is

near an amphidrome in the semi-diurnal tide. In fact, Southampton lies close to the nodal line for the semi-diurnal tide in the English Channel. Taking another example, in the Irish Sea, there is a degenerate amphidrome for the semi-diurnal tide close to Courtown on the Irish coast. Here, the semi-diurnal tide is small and the higher harmonics can be the dominant components in the tide Pugh (1987). Alternatively, in places where the two principal semi-diurnal constituents – those of the sun and moon – are about equal, the D2 tide will become small twice a month at neap tides and any higher frequency oscillations will assert

themselves. In South Australia, this phenomenon is called the dodge tide Nunes and Lennon (1986).

These considerations about small values of a lead us to the conclusion that there must be a second criterion for double high waters. As $b/a$ increases from the critical value given by Eq. (1), a double high water will first form and then, as $b/a$ continues to increase, the double high water will disappear as the higher harmonic becomes dominant. The Doodson criterion in Eq. (1) for the production of double high tides is therefore a necessary requirement but it is not, on its own, a sufficient requirement.

We need also to impose an upper limit on the ratio $b/a$ to allow for times and places where the semi-diurnal tide is small compared to its harmonics. We can imagine that a more general condition for double high waters might take the form

$$x > \frac{bn^2}{a} > 1 \tag{2}$$

where $x$ is a function to be determined. Our first aim in this paper is thus to explore the nature of $x$.





A further restriction on the Doodson criterion is that it applies only to the case where the phase of the higher harmonic is optimal: that is, in the case of double high waters the minimum in the harmonic coincides with the maximum in the semi-diurnal tide. Bowers et al. (2013) suggested a modification to the Doodson criterion to allow for a phase difference between the main tide and its harmonic. A second aim of this paper is to take the opportunity to test that idea against observations.

## 2 Theoretical background

Figure 1 illustrates the formation and subsequent disappearance of a double high water for the sum of a semi-diurnal D2 tide (amplitude $a$) and its D6 harmonic (amplitude $b$). In Fig. 1a the ratio $b/a = 0.25$ and the phase difference is 0.5 hours (i.e., the minimum in D6 occurs half an hour after the maximum in D2). Adding a D6 curve to D2 creates a double high water and also a double low water, as the maximum in D6 also coincides with the low water in D2. In Fig. 1b the amplitude of D2 has been reduced to 0.1m, keeping the D6 curve the same. There is now no evidence of a double high water: the tide is instead best described as 6th diurnal with a weak remnant semi-diurnal modulation in mean water level.

In general the sum of a semi-diurnal tide and a single higher harmonic can be written

$$y = a\cos[\omega t] - b\cos[n\omega(t - \phi)] \tag{3}$$

where $y$ is sea level and $\omega = 2\pi/12$ lunar hours is the angular frequency of the semi-diurnal tide. The phase $\phi$ represents the time difference between the maximum in the semi-diurnal tide and the nearest minimum in the higher harmonic. Time, $t$, is measured from zero at the high water in the semi-diurnal tide. For a small range of times around $t = 0$ and $t = \phi$ the cosine curves can be represented by quadratic curves (using the expansion that $\cos(x) = 1 - 0.5x^2$ for small $x$):

$$y = a[1 - 0.5\omega^2 t^2] - b[1 - 0.5n^2\omega^2(t - \phi)^2] \tag{4}$$

When there is a double high water, there is a turning point (marked as D on Fig. 1a) which marks the centre of the dip in sea level between the two high waters either side. At this turning point, $dy/dt = 0$ and so, for small values of $\phi$, from Eq. 4,

$$t' = \frac{n^2(b/a)}{n^2(b/a) - 1} \tag{5}$$

Since the term in brackets is always greater than 1, $t' > \phi$. It is a feature of double high waters that the turning point lies further away from the high water than does the minimum in the harmonic that creates it.

For the dip to be a minimum between the double high waters, a further condition is that $d^2y/dt^2 > 0$ at the dip. At this point it is necessary to include the next term (a function of $(t - \phi)^4$) in the expansion of the cosine curve for the higher harmonic. Applying the condition that $d^2y/dt^2|_{t=t'} > 0$ gives the condition for a double high water as

$$\frac{b}{b}r^2 > 1 \tag{6}$$

where

$$r^2 = n^2[1 - (0.5n^2)\omega^2(t' - \phi)^2] \tag{7}$$





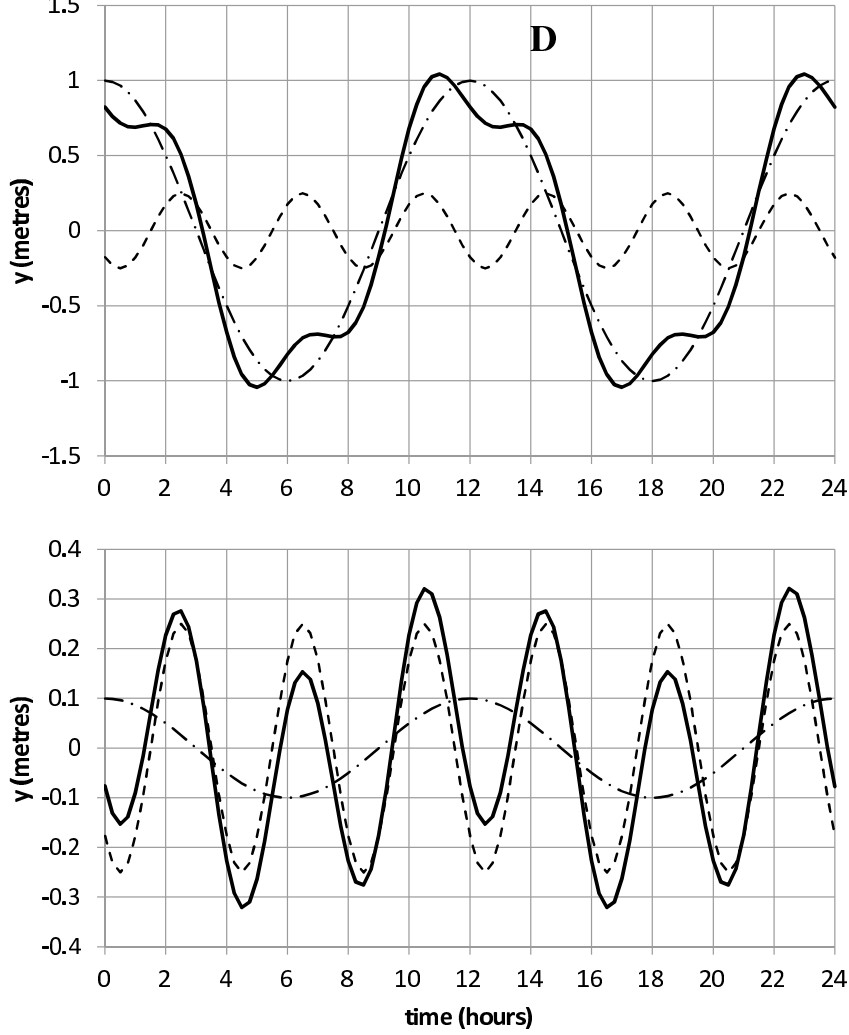

**Figure 1.** a) A double high (and double low) water formed by the addition of a semi-diurnal curve (dash-dot line, amplitude 1m) and a sixth-diurnal harmonic (dashed line, amplitude 0.25m). There is phase difference of 0.5 hours between the curves. D marks the centre of the dip between the high waters. At D, the gradient of y with respect to time is zero.

b) As a) but the amplitude of the semi-diurnal curve is reduced to 0.1 metres. The double high and low waters have disappeared and the tide is effectively sixth diurnal.

which is the result obtained by Bowers et al. (2013). Note that when the phase difference $\phi=0$ it follows that $t'=0$ and therefore that $r^2 = n^2$: the condition for a double high water then becomes the Doodson criterion in eq. 1. For all other values of $\phi$, $r^2 < n^2$; then, according to Eq. (6), $b/a$ must be larger than the value required for zero $\phi$ in order to create a double high





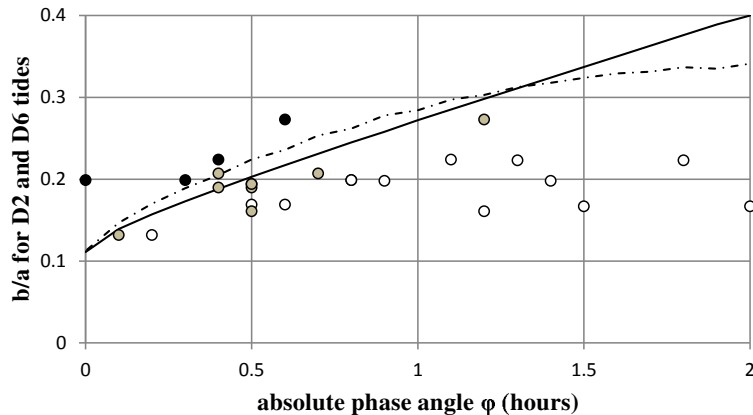

**Figure 2.** Theoretical condition for the formation of double high waters illustrated for the addition of a sixth diurnal (D6) harmonic to a semi-diurnal D2 tide. The y-axis is the ratio of the amplitudes of the harmonic to the principal tide. The x-axis is the time difference between the minimum in D6 and the maximum in D2(plus D1 in the case of the observations). The solid curve shows the critical value of $b/a$ for the formation of double high waters according to the analytical solution (Eq. (6) in text). The dotted curve is an exact numerical solution to the problem. The points show values of $b/a$ and $\phi$ at Port Ellen. Points have been plotted as solid circles for tides where there is a double high water, grey filled circles where there is a stand in the tide near high water and open circles when no stand nor a double high water is observed.

water. It is possible for $r^2$, and hence $(b/a)r^2$, to be negative. This will happen when either $b/a$ is small or $\phi$ is large. When $r^2$ is negative, it is impossible to satisfy Eq. (6) whatever the value of $b/a$, and no double high water can be formed.

As an illustration, we show in Fig. 2 the solution to Eqs. (4)–(6) for the case of $n = 3$, that is the sum of D2 and D6 tides. The solution is symmetrical for negative and positive values of f and so in Fig. 2 we have drawn the curves as a function of the absolute value of $\phi$. It is necessary to iterate to reach the solution: the critical value of b/a must satisfy both Eqs. (5) and (6). For D2 and D6 tides, the critical value of $b/a$ for zero phase is 1/32, or 0.111. As the phase difference increases from zero, the critical value of $b/a$ also increases, so that for a phase difference of 1 hour, it is necessary for the amplitude of D6 to be at least about 0.27 times that of D2 to produce a double high tide. Also shown on Fig. 2 is the exact numerical solution to the problem, starting with the cosine curves in Eq. (3). The approximation of the cosine curves as quadratic curves close to their maxima and minima means that the analytical solution is not exact, but it does capture the essential features of the numerical solution, especially for $\phi$ less than about 1.5 hours. In fact, at values of $|\phi|$ greater than about 1 hour the dip becomes so detached from the high water that the tidal curve is no longer recognisable as a double high water. In practice, therefore, for a recognisable





double high water produced by D2 and D6 tides, we can limit attention to the region in which $\phi$ is less than about 1 hour on Fig. 2.

As the amplitude ratio $b/a$ continues to increase above the critical value for the formation of a double high water, there is a gradual transition towards a tide dominated by the higher harmonic. As this happens, the level of water in the dip between the high waters falls towards the level of the low tide (Fig. 1). There comes a point when the level of water between the dips is virtually the same as that at the low tide, when we can say that the transition to the higher harmonic is complete and the double tide has disappeared. At the time of the minimum in the dip, $t = t'$ and sea level is given by

$$y = a(1 - 0.5\omega^2 t^2) - b[1 - (0.5n^2)\omega^2(t - \phi)^2] \tag{8}$$

The first term on the right hand side of this equation represents the fall in water level between $t = 0$ and $t = t'$ due to the semi-diurnal tide; the second term is the fall in the same time due to the harmonic. The first term will occur in the absence of the harmonic, so the size of the dip due to the harmonic is equal to the second term. Referring to Eq. (7), this second term can be written as $b(r/n)^2$. The size of the dip relative to the amplitude of the semi-diurnal tide is therefore $(b/a)(r/n)^2$. The higher harmonic will become dominant when this ratio exceeds a certain value, of order 1. That is the upper limit of $b/a$ for a recognisable double high water is

$$\frac{n^2}{f} > \frac{b}{a} r^2 \tag{9}$$

where the factor $f$ represents the size of the dip relative to $a$. For example, if $f = 1$ and $n = 3$, the upper bound for $b/a$ is $9/r^2$. In general, we can write the critical condition for the formation of a recognisable double high water in the form

$$\frac{n^2}{f} > \frac{b}{a} r^2 > 1 \tag{10}$$

which is the condition we want to test. Note that the theory leads us to expect that the same parameter, namely $(b/a)r^2$, is important in predicting the initial onset of the double high water and the disappearance of the high water as the harmonic becomes dominant.

## 3  Observations

Port Ellen lies on the south coast of Islay, an island which is part of the Inner Hebrides on the west coast of Scotland (Fig. 3). Tidal data at Port Ellen are available from the Permanent Service for Mean Sea Level (PSMSL; http://www.psmsl.org/), and Fig. 4 shows a record from Port Ellen for the second half of February, 2010. The 2-week observational period we use began on a new and ended on a full moon. The semi-diurnal tide is unusual because the two principal semi-diurnal tidal constituents (M2 and S2) are about equal in amplitude and, as a consequence, the semi-diurnal tide virtually disappears twice each month when these two constituents are 1800 out of phase. This effect can be seen on the 25th of the month in Fig. 4. A similar phenomenon is the dodge tide in South Australia (Nunes and Lennon, 1986).





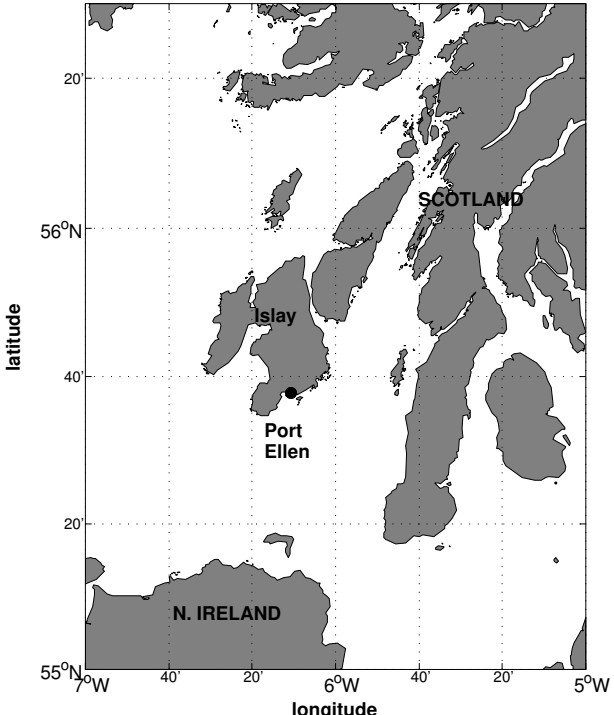

**Figure 3.** Location of Port Ellen in the Inner Hebrides, Scotland

The diurnal tide is also important at Port Ellen and produces a significant diurnal inequality which can be seen at the beginning and end of the record in Fig. 4. In the figure, the diurnal inequality takes the form of the low tide in the middle of the day being higher than the low tide at the beginning and end of the day. The diurnal inequality is important in the selection of the morning or afternoon tide for the production of double high waters, as we shall see.

The double high water is intermittent and occurs most clearly in the morning high tide in the first part of the record (the 18th and 19th) and the afternoon high tide towards the end of the record (the 27th and 28th). There are also tides when there is a stand in water level around the time of high water: it is sometimes difficult to tell if there is actually a double high tide present on these occasions or not. These 'stand' tides are observed on all tidal cycles between the morning of the 15th and the morning of the 17th, and on the mornings of the 20th, 23rd and 27th.

## 3.1 Harmonic analysis

We have analysed short portions of the water level record for the amplitude and phases of key harmonics using harmonic analysis (see, e.g., Emery and Thomson, 1996, for details). The method was first described by Airy (1843) to establish the fact that higher harmonics of the semi-diurnal tide were important at Southampton. Later, Doodson and Warburg (1941) used the technique to test their inequality in Eq. (1) against observations of the tide at Southampton.



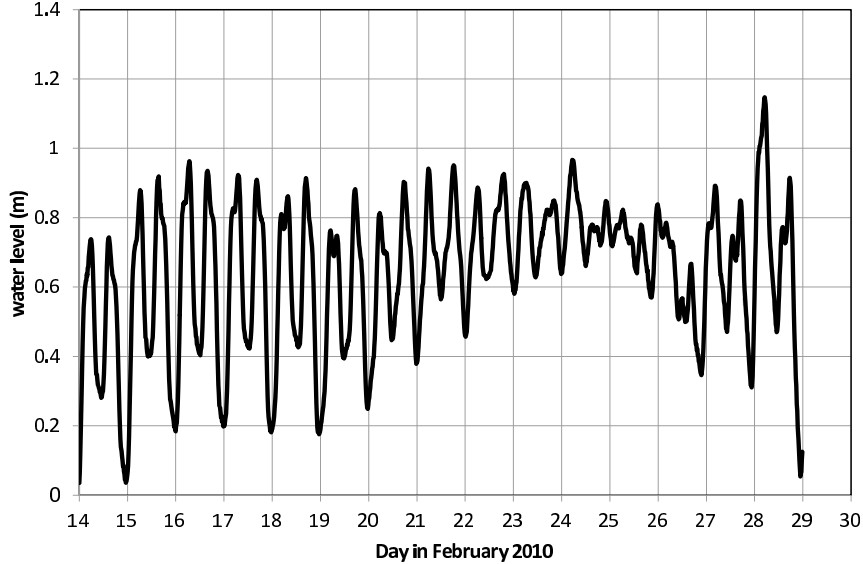

**Figure 4.** Water level record for Port Ellen in the second half of February, 2010. Double high waters are seen on the mornings of the 18th and 19th and the afternoons of the 27th and the 28th of the month.

Because the diurnal tide is important at Port Ellen, we have analysed the data for harmonics starting with the diurnal tide as the first harmonic. The data set of Fig. 4 was broken up into 25 hour segments, each starting at midnight on the chosen day. The selected segments of data therefore run for one hour into the next day. The harmonic analysis is then applied to each segment of data to calculate the amplitude and phase of a diurnal constituent (D1, period 25 hours), a semi-diurnal constituent

5 (D2, period 12.5 hours), a quarter-diurnal constituent, D4, period 6.25 hours) and a sixth-diurnal constituent (D6, period 4.17 hours). Note that this analysis produces amplitudes and phases of the harmonics applicable to that day only. The harmonics are therefore not the same as the tidal constituents such as M2 and M4 treated by tidal analysis and which have constant amplitude and phase on all days. The amplitude and phase of these daily harmonics changes from day to day. The usefulness of the 'D' harmonics has been discussed by Pugh (1987). It enables the relative importance of the principal tide and its harmonics to be

10 established on each day. The relationship between the relative amplitudes and phases of the harmonics and the production (or non-production) of double high waters can then be explored.

Figure 5 shows an example of the curve-fitting for the 19th of February: Fig. 5a showing the observed and fitted curve for this day. The double high water on the morning tide is clearly shown and the fitted curve reproduces this. There is no double high water in the afternoon, in either the observed or fitted tide. The fitted curve is not perfect, however. Most significantly, there

15 is a double low water in the fitted curve, which is not seen in the observations. Figure 5b shows the nature of the harmonics which sum to give the fitted curve on this day. The semi-diurnal harmonic, D2, has the largest amplitude, followed by D1, then D6 and finally D4. The low water in the D6 curve is close to the high water in the D2 curve, a requirement for the formation





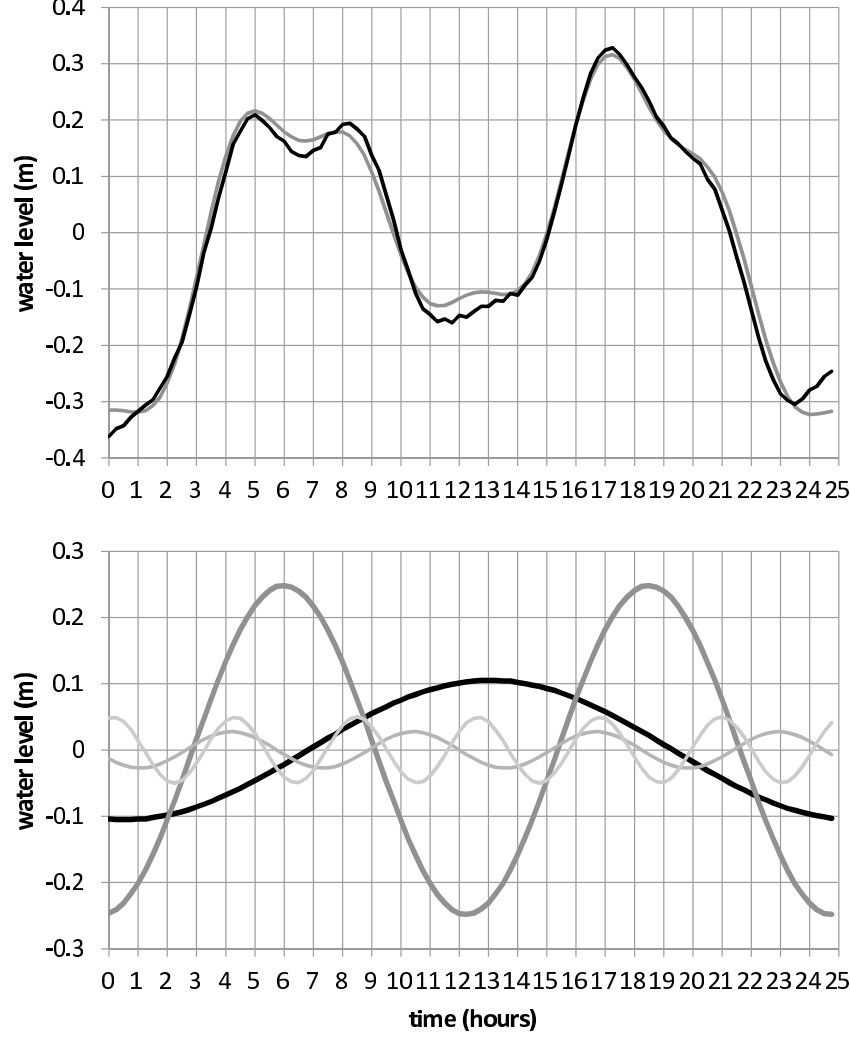

**Figure 5.** a) Observed and fitted tidal curves for 19th February. The double high water on the morning is reproduced by the fitted curve.

b) D1, D2, D4, and D6 harmonics for 19th February (note that the higher the harmonic, the lighter the line colour). The high tide in D1 (black line) occurs in the middle of the day and causes the diurnal inequality seen in Fig. 3a. The D6 harmonic (lightest grey) has a larger amplitude than D4 and the minimum in D6 occurs closer to the high D2 than does the minimum in D4. It is the D6 harmonic that is primarily responsible for the double high water at Port Ellen.

of double high waters. In contrast, the amplitude of D4 is smaller than D6 and its phase is not optimal: the low in D4 occurs about 2 hours after the semi-diurnal high water in Fig. 5b.

The fact that there is a double high water in the morning of the 19th, but not in the afternoon, can be explained by the effect of the D1 tide. The diurnal tide is rising in the morning and falling in the afternoon (Fig. 5b). This has the effect of pushing





the time of high water forwards in time, towards noon, in the morning and dragging it backwards in time, also towards noon, in the afternoon. The time interval between the two high waters produced by D2 and D1 together is therefore less than 12.5 hours. It is therefore possible for the low tide in D6 to be close to the high tide in D1 and D2 in the morning, but not so close in the afternoon. In fact, the time interval between the D6 low and the (D1+D2) high on the 19th is zero in the morning and 0.8

hours in the afternoon. The close coincidence of the low water in D6 and the high water in (D1+D2) produces a double high water in the morning but not in the afternoon. For this reason, it is important to consider the time of the high water in D1 and D2 combined.

The Fourier analysis illustrated in Fig. 5 has been applied to each of the days in the record and the results which we shall use in the analysis are shown in table 1. The phase of D6 is expressed as the absolute value of the time interval between low water

in this harmonic and the nearest high water in D1 and D2 combined. The semi-diurnal tide virtually disappears on the 24th, at neap tides, because of the equality in the amplitudes of the solar and lunar tides.

### 3.2 Testing the condition for double high waters

A full analysis of the conditions that produce a double high water would include all of the relevant harmonics of the semi-diurnal tide. In the case of the short record at Port Ellen, however, the mean amplitude of D4 is 0.026m and that of D6 is 0.044m.

The fact that D6 is generally larger, coupled with the fact that it is of higher frequency and so more potent at producing double high tides, means that, if we are to consider just a single harmonic, then that should be D6. Accordingly, we limit our analysis in this section to the production of double high waters by a combination of D6 and D2 tides, and use this combination to test the theory of section 2. We bear in mind, however, that the comparison between observations and theory may not be exact because we are neglecting other harmonics, notably D4 and also D1. It can be seen in table 1 that on all days the ratio of amplitudes $b/a$

for D6 and D2 tides is greater than the value of 0.111 required to satisfy the condition (1). Moreover, the value of $b/a$ is nearly as great on the afternoon of the 24th (0.223, when no double high water is seen) as it is on the afternoon of the 28th (0.224, when there is a double high water). The straightforward Doodson criterion, neglecting the effect of phase difference between the harmonic and the principal tide, is therefore not the best discriminator for double high waters. The rows in table 1 have been marked for tides where a double high water is observed, or nearly so. Tides in which a double high water is observed are

shaded in the darker grey and those where there is a stand in the tide, so that a double high water is close to being formed are marked in lighter grey. On days when there is a clear double high water, that is on the morning of the 18th and 19th and on the afternoons of the 27th and 28th, the value of $(b/a)r^2$ is greater than or equal to 1.5. On days when there is a stand in the tide close to high water, $(b/a)r^2$ is greater than about 0.5 and less than 1.5. On all other days, when there is no double high water (excepting the 25th and 26th) the value of $(b/a)r^2$ is less than 0.5. The amended Doodson condition therefore performs well

as a predictor of double tides, although the transitions are not as clear-cut in the observations as theory would predict. This is likely to be partly because of difficulty in deciding when a double high water is there by visual inspection alone and partly because the real Port Ellen tide has harmonics other than the D2 and D6 components treated in the theory.

On the 25th and 26th of the month, $(b/a)r^2$ reaches its highest values, over 11 on the 25th and a more modest level of over 2 on the 26th. There is no clear double high water on these days, though. Instead the tide is confused with no clear semi-diurnal





pattern (Fig. 4). Closer inspection of the tides on these days (using plots equivalent to Fig. 5) shows that on the 25th the diurnal and quarter diurnal tides are dominant: There is certainly no double high water in the semi-diurnal tide on this day. The situation on the 26th is more complicated. The diurnal tide is the largest component on this day. If the diurnal tide is removed from the reconstruction and a tidal curve is drawn as the sum of the D2, D4 and D6 harmonics, then a double high water can

be seen in both the morning and afternoon. The predictor is therefore doing a reasonable job, within its limitations, throughout the fortnight of the observations. When the parameter $(b/a)r^2$ has a value less than about 1, no double high water is formed because the higher harmonic is too weak. When it has a value greater than about 10, then again no double high water is formed because the higher harmonic is too strong. Between these limiting values, double high waters, or at least a tidal stand, are likely to be observed, provided they are not obscured by the presence of other harmonics.

We have plotted, on Fig. 2 values of $b/a$ for the D2 and D6 tides (from table 1) against the phase difference $\phi$ (also from table 1). The points have been coded so that those on tides in which a double high tide is observed are shown as filled circles and those on which no double high tide is observed as open circles. Tides where there is a point of inflection, or a stand in the tide, have been marked as grey-filled circles. The tides of the 25th and 26th of the month, when the semi-diurnal tide is very small, have been omitted from this diagram. We expect double high waters to occur when the points lie above the theoretical curves

on this diagram and this is generally the case. All four clear double high waters plot above the theoretical transition curve. Similarly, points representing the tides with no double high water lie below the critical curve. The grey points, representing tides with a stand, lie close to the critical lines. It is illuminating to note that the main spread of the points in Fig. 2 is along the x-axis. Because the amplitude of harmonics generated by the semi-diurnal tide tend to increase with that of D2, the ratio $b/a$ remains fairly constant. The critical factor in deciding whether a double high tide will form is actually the phase difference. As

the time of the dip in D6 moves close to the high tide in D2, the critical condition for a double high tide is met. For this data set, therefore, and perhaps in general, the phase difference between the harmonics is the most important parameter in controlling the formation of a double tide.

## 4   Discussion

The theoretical considerations presented in this paper, supported by a small data set, suggest that a single parameter can be used

to predict the presence of a double high water when a higher harmonic is added to a semi-diurnal tide. As we might expect, this parameter – $(b/a)r^2$ – depends on the amplitude and phase of the harmonic (compared to the semi-diurnal tide) and on the ratio of frequencies of harmonic and main tide. The data in table 1 can be divided into 4 categories:

$(b/a)r^2 \leq 0$:no double high water is seen

$0.5 \leq (b/a)r^2 \leq\sim 1.5$:a stand in the tide is observed, but no clear double high water

$1.5 \leq (b/a)r^2 \leq\sim 10$:the regime of double high waters

$(b/a)r^2 > 10$:the harmonic dominates and there is again no clear double high tide





In the terms posed in the introduction, it is necessary to place both lower and upper bounds on the criterion for a double high water to allow for the higher harmonic being too small and too large.

A limitation of the theory, as presented here, is that it considers just a single higher harmonic added to the semi-diurnal tide. There will be places, and times, when this is appropriate. In the case of the data from Port Ellen we present here, the theory

adequately represents the data for most of the time. The theory can be extended to include other harmonics in a straightforward way. The condition for the initial formation of a double high water by two harmonics added to the semi-diurnal tide, for example, can be written:

$$\frac{b_1}{a}r_1^2 + \frac{b_2}{a}r_2^2 > 1 \tag{11}$$

Here, $b_1$ and $b_2$ are the amplitudes of the two harmonics and $r_1^2$ and $r_2^2$ are calculated from Eq. (7) with the appropriate

phases. The time at the centre of the dip, $t'$, is calculated from a modified form of Eq. (5) which allows for both harmonics acting together. It can happen that the two harmonics support each other in producing a double high water or (because the $r_2$ parameter can be negative) that one of the harmonics suppresses a double high water that would otherwise be formed by a single harmonic acting on its own.

Turning now to the observations, it is, in fact, difficult to tell, in marginal cases, when there is a double high water by visual

inspection alone. This is because, when the double high first forms, the dip between the two high waters is small – perhaps just a centimetre or so, and this is often at the noise level of the measurements. In this paper, we have tried to avoid this problem by referring to a tidal stand – that is, there is clearly a portion of the tidal record where the water level remains flat for a time, but it is difficult to say for sure if there is a double high tide. A useful goal for future studies in this field would be to formalise a parameter for defining the presence and the magnitude of double tides.

For most of the data set, the two most important harmonics at Port Ellen are the semi-diurnal (D2) and the sixth-diurnal D6. Other harmonics, in particular D1 and D4 play a role, however. The effect of the diurnal harmonic, D1, in allowing a double high water in one tide but not in the second on the same day has already been mentioned. The role of the D4 harmonic at Port Ellen is to suppress the formation of the double low tide. The sum of D2 and D6 harmonics tends to produce a double high tide and a double low tide (Fig. 1a). At Port Ellen, the minimum in D4 occurs at about the time of the minimum in D2 (see

Fig. 5b). The D4 harmonic, though small, tends to flatten out the low tide and prevents the sixth diurnal harmonic (which has a maximum at this time) from producing a double low water. The opposite is true at high tide, when the minimum in D4 helps the minimum in D6 to produce a double high water.

## 5    Conclusions

The formation of double high and double low waters in the semi-diurnal tide is a fascinating problem but it is not a big topic

in tidal studies because it happens at just a few locations. The reasoning we have applied in this paper, however, is relevant to a much more general problem: the classification of ocean tides. The most basic classification of ocean tides is in terms of their principal period: diurnal, semi-diurnal or a mixture of these two (called mixed tides). The type of tide is often characterised





quantitatively by a form factor, which is the ratio of the amplitudes of the main semi-diurnal constituents (M2 and S2) to the main diurnal constituents (O1 and K1). Each port can be given a form factor depending on this ratio and the type of tide to be expected can then be gauged from the form factor. The drawback with this classification system is that the relative importance of semi-diurnal and diurnal tides at a port can change from day to day with the springs-neaps fortnightly cycle. The type of tide

5    at a port can therefore also change, for example, from semi-diurnal to diurnal and back again over a fortnight. This is exactly the same problem that we have been considering in this paper, only with D1 and D2 tides rather than the D2 and D6 we have considered. Although we will expand no further on the topic here, the analysis we have described in this paper could easily be adapted for characterising the temporal changes in the nature of the tide at a port.

*Competing interests.* The authors declare that they have no conflict of interest.

10   *Acknowledgements.* Funding was provided by the Natural Environmental Research Council (grant NE/I030224/1 to JAMG), and from the BurningHam research support foundation.



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





**Table 1.** The first column shows the day in February 2010 divided into two tides, one in the morning (am) and a second in the afternoon (pm). $a$ is the amplitude of the semi-diurnal tide in metres and is the same in the morning and in the afternoon. $b/a$ is the ratio of the amplitudes of the sixth diurnal tide D6 to that of D2. $|\phi|$ is the absolute value of the time difference between the maximum in D1 and D2 combined and the minimum in D6; $t'$ is the time interval calculated with Eq. (5), and $r^2$ is the parameter in Eq. (7). Finally, $(b/a)r^2$ is the discriminator for double high waters. Tides in which a double high water is observed are shaded in the darker grey and those where there is a stand in the tide, so that a double high water is close to being formed are marked in lighter grey.

| Tide | $a$ | $b/a$ | $|\phi|$ | $t'$ | $r^2$ | $(b/a)r^2$ |
|------|-----|-------|----------|------|-------|------------|
| 14am | 0.28 | 0.169 | 0.6 | 1.75 | -4.57 | -0.77 |
| 14pm | 0.28 | 0.169 | 0.5 | 1.46 | -0.43 | -0.07 |
| 15am | 0.29 | 0.207 | 0.7 | 1.51 | 2.27 | 0.47 |
| 15pm | 0.29 | 0.207 | 0.4 | 0.86 | 6.8 | 1.41 |
| 16am | 0.32 | 0.19 | 0.5 | 1.2 | 3.92 | 0.75 |
| 16pm | 0.32 | 0.19 | 0.4 | 0.96 | 5.75 | 1.09 |
| 17am | 0.31 | 0.194 | 0.5 | 1.17 | 4.4 | 0.85 |
| 17pm | 0.31 | 0.194 | 0.5 | 1.17 | 4.4 | 0.85 |
| 18am | 0.29 | 0.199 | 0.3 | 0.68 | 7.53 | 1.5 |
| 18pm | 0.29 | 0.199 | 0.8 | 1.81 | -1.47 | -0.29 |
| 19am | 0.25 | 0.199 | 0 | 0 | 9 | 1.79 |
| 19pm | 0.25 | 0.199 | 0.8 | 1.81 | -1.47 | -0.29 |
| 20am | 0.21 | 0.161 | 0.5 | 1.61 | -3.69 | -0.59 |
| 20pm | 0.21 | 0.161 | 1.2 | 3.87 | -64.1 | -10.3 |
| 21am | 0.19 | 0.167 | 1.5 | 4.48 | -82 | -13.7 |
| 21pm | 0.19 | 0.167 | 2 | 5.98 | -152 | -25.5 |
| 22am | 0.15 | 0.198 | 1.4 | 3.19 | -23.8 | -4.71 |
| 22pm | 0.15 | 0.198 | 0.9 | 2.05 | -4.56 | -0.9 |
| 23am | 0.12 | 0.132 | 0.1 | 0.63 | 6.1 | 0.81 |
| 23pm | 0.12 | 0.132 | 0.2 | 1.26 | -2.58 | -0.34 |
| 24am | 0.07 | 0.223 | 1.8 | 3.59 | -23.7 | -5.28 |
| 24pm | 0.07 | 0.223 | 1.3 | 2.59 | -8.06 | -1.8 |
| 25am | 0.02 | 1.29 | 1.3 | 1.42 | 8.85 | 11.43 |
| 25pm | 0.02 | 1.29 | 0 | 0 | 9 | 11.63 |
| 26am | 0.1 | 0.399 | 1.6 | 2.22 | 5.1 | 2.03 |
| 26pm | 0.1 | 0.399 | 0.2 | 0.28 | 8.94 | 3.6 |
| 27am | 0.19 | 0.273 | 1.2 | 2.02 | 2.06 | 0.56 |
| 27pm | 0.19 | 0.273 | 0.6 | 1.01 | 7.26 | 1.98 |
| 28am | 0.36 | 0.224 | 1.1 | 2.18 | -3 | -0.67 |
| 28pm | 0.36 | 0.224 | 0.4 | 0.79 | 7.41 | 1.66 |