# Peer review of "The double high tide at Port Ellen: Doodson's criterion revisited"

_Ocean Science, 2017_

## Referee Comment (RC1) · Anonymous Referee #1 · 19 Apr 2017

General

This is a well written and appropriately brief account of how tidal harmonic constituents may add to give double high or low waters. The authors extend the simple Doodson criteria to cases of non-optimal phase relationships, and illustrate the theory with an example from Port Ellen in the Scottish Inner Hebrides.

Summing harmonics is essentially an exercise in trigonometry, rather than an investigation of tidal Kelvin wave dynamics. A regional study of wave generation and propagation around the amphidrome would be potentially more enlightening. Double high waters occur in a very few places worldwide, almost always associated with tidal amphidromes, where the range is near to zero. As such, as the authors state, this is "not a big topic" in tidal studies. Nevertheless, their extension of Doodson's simple formula

does take it a stage further, and also shows how difficult and unrewarding ever further elaboration of the criteria through the trigonometry approach becomes; the paper should certainly be published for this reason, as well as for the intrinsic analysis.

Specific comments.

Line 1 Doodson's was a minimum criterion.

L12 here "neap" is used for the time of minimum local tidal ranges, though because of amphidrome movements during the spring-neap cycle, this may not be near first and last lunar quadrature when small neap ranges occur in the tidal forcing.

L25 "correct" is not the right word. Maybe "an appropriate" is better.

L1 a third reason for higher harmonics is streamline curvature around indented coasts and sansbanks.. Godin is a very obscure reference for such a universal truth.

L10 replace "stringent" with "minimum"

L 13 explain why phi has a negative sign

Bottom line be consistent in Eq. or eq. there may be other examples.

Caption last line is incorrect grammar ..should be " when neither a stand, nor a double..."

L 6 1/9 no1 1/32

L5 an enlarged plot of selected days would help. Fig 1 is a bit too compressed.

L24. Say what is the sampling interval (15 minutes) and correct the source specification. It is not PSMSL. PSMSL is an international body located at NOCL which publishes monthly and annual mean sea levels for IAPSO. UK 15-minute sea level data comes from BODC (also at NOC Liverpool) under contract from DEFRA, SEPA etc. There is a specific form of acknowledgement required...see their web site. As DEFRA through the Environment Agency, and SEPA pay a lot of money for this measurement programme they like to get some credit.

L25 show the amphidrome in Figure 4

L27 this is similar to Courtown tides and worth a brief comparison (see Pugh 1982)

L28 180 degrees not 1800.

L10 here the authors seem to be calling "Harmonic analysis" narrowly the fitting of daily curves. D and W actually used Y1, Y2..etc. Say so.

L3 If they are 15-minute readings, a better symmetry would have been to go from 2330 in day -1 to 0030 on day +1.

L7 "treated by a full harmonic tidal analysis.."

L11 A plot of each of the daily D1, D2, D4 and D6 amplitudes and a discussion would be useful here to understand what is happening.

L14 Note that observations include met effects, seiches and instrument noise.

The line darkening doesn't work well. Surely web publication allows full colour?

L8 HA seems now to be called Fourier Analysis. Non-mathematical readers may find this alternative naming confusing. Be consistent.

L15 specifically the advantage for D6 is $(9/4)*(44/26)$...say so.

L 32 and noise from Met, seiching etc..

L18 This is an important discussion and it would be useful to plot the D4 and D6 amplitudes against D2 here. See Pugh and Woodworth 2014 for what happens at Southampton. This interdependence of the basic and higher harmonics has big influences on the turning points occurrences.

L21 not sure why phase is said to be of "most importance"...it's both amplitudes and phases..."perhaps in general" is just too vague and unjustified.

L8 Doodson and Warburg discuss the "three plus harmonics" case, on page 222, for amplitudes only.

L21 this just confirms how unrewarding further complicating the criteria would be.

L1 and 2 Replace the two "and"s in parenthesis with + . It's in the definition.

The suggestion that the designation semidiurnal, mixed and diurnal needs redefining to allow for daily changes is valid, but a needless complication. The whole point about this basic descriptor is its simplicity. As an example of how a time dependent Form Factor might go, at lunar zero declination the D1 is near zero, and so the regime form factor ratio could be zero globally (ok, ignoring solar declination), but how would that

improve our simple descriptor of local tides at a port?

In Table 1 the values of R**2 have negative values sometimes. Is it worth considering the implied virtual values of r? might be worth a comment on page 3.

---

## Referee Comment (RC2) · Anonymous Referee #2 · 28 Apr 2017

THIS IS NOT A REVIEW. SO PLEASE IGNORE THE MANUSCRIPT RATINGS, SINCE THIS WEBSITE REFUSES TO WORK UNLESS THOSE BUTTONS ARE CLICKED.

When I began reading this paper, I was inclined to send in a favourable review. However, when I tried checking the mathematics, I ran into so many careless errors that I finally gave up. Please have the authors revise their manuscript and check it carefully this time. After that, I will then review the paper.

Specifically, they need to address the following:

Equation (5) does not follow from Eq (4).

The line after Eq (5) refers to a term in brackets. What brackets?

[Figure]

Eq (6) is clearly wrong.

Page 5, line 4, a new variable "f" is introduced but not defined.

Figure 1 has a floating label "D" but it doesn't point to anything.

In their revision, the authors may also wish to consider a previous paper by Godin, published in 1993 in the Deut Hydrogr Zeitschrift.

---

## Referee Comment (RC3) · P.L. Woodworth (Referee) · 30 Apr 2017

30 April 2017

Comments on "The double high tide at Port Ellen: Doodson's criterion revisited" by Byrne et al. (OSD)

This is an entertaining little paper in the style of papers by tidal scientists a century ago. It discusses the 'Doodson criterion' whereby harmonics (M4, M6 etc.) of the main tidal constituent (M2) can generate a double high or low water such as occurs at Southampton. The criterion, which is mentioned in Doodson and Warburg (1941, p.221) and more recently in Pugh and Woodworth (2014, p.138), is clearly stated to reflect the ideal situation of producing a double tide, when the phase relationships are optimal. The present paper aims to extend consideration of that special situation

into the more general case when there are different relative amplitudes and phases between the different constituents.

I found it to be quite an educational exercise and it overlaps with some of my own work as will be apparent from the comments below. So I have no objection to its publication. I have some small comments and a couple of more general ones.

Small comments:

p1, first para of the Introduction -I found this wording disappointing as the authors have just paraphrased p133 of Pugh and Woodworth (2014). I think if I had been them I would have worded it to say that, given the spatial scales of the harmonics are shorter than that of M2 (as they discuss below), then whenever you have a place like S'ton with a double high then there is probably a place with a double low a short distance away (Weymouth in this case). For example, if you travel along the French/Belgian/Dutch coast, double highs are interleaved with double lows several times, not just at isolated places like Den Helder. This becomes most evident using tide models of course.

Then some mention of other places such as the US would be appropriate by all means. There is discussion of Weymouth and the Doodson criterion in Woodworth (2017). The Redfield reference for New England corresponds to what Woodworth (2017) calls the 'Newport enclave'.

Finally, I suspect there may be a location not too far from Port Ellen where there are double lows at times. Maybe you could look into that.

p2, 14 - value of it

same line - spring-neap

19 - I wouuld say 'about 4 hours'

25 (Nunes and Lennon, 1986)

26 - a should be in italics

28 - This wording might confuse some people. It is quite reasonable for you to investigate when the harmonics are larger than M2, and it is an extension to Doodson's work, but it is not an extension to the Doodson criterion as such, which was explictly to do with harmonics smaller than M2.

p3, 7 - b/a = 0.25 and a = 1 metre

21 - you should state that tprime is the time of the dip

23 - I think this should read 'in the (semidiurnal) high water'

eq.6 - b/b should be b/a

p4, fig.1a - the D seems to at 14 hours and not at the time of the dip at about 12.5 hours. Can you have an arrow pointing to the dip?

In the caption, mention that time is measured from the peak of the M2 tide, as you do in the text.

p5, 4 - f should be phi

6 - 1/32 should be 1/9

7-8 - 'at least about' sounds like you don't know what it really is. drop 'about'.

I had to read this paragraph twice as it was not clear at the start that what was being discussed was the approximate parameterisation - it became clear at line 9, and it is clear in fig.2 caption. Perhaps review the wording here.

p6, 3 - I wondered if a subheading was appropriate here as the discussion now switches to the opposite situation of large b/a.

section 3 - I don't know where you got the data from - it may well have been from someone in PSMSL, but PSMSL is not usually associated with the UK 15-minute data. I would credit the source as BODC (www.bodc.ac.uk) or NTSLF (www.ntslf.org), or maybe you got it from the IOC Facility (www.ioc-sealevelmonitoring.org).

28 - 1800 should be 180 deg

drop the last sentence this page - this repeats from above.

p7, Fig.3 - I would drop this. The exact location of Port Ellen is of no importance to the reader, all he needs to know is that it is a place with similar M2 and S2 amplitudes. If you want to keep it, I would make it more useful by adding the M2 tidal map from a model - I am sure Dr. Green has one.

6 - tides should be times

Fig5 caption - Observed (bold black) and fitted (grey) ...

line 3 - there is no Fig.3a. You mean 4 or 5a?

p11, 14-17 from 'We expect' to 'critical lines'. Why do you expect this etc? It is not obvious to me.

24-31 - make it clear that these limits derived from table 1 apply to an M2/M6 situation and not an M2/M4 situation.

p12, 18-19 - there are already criteria for defining double tides in some operational agencies e.g. in the US (NOAA), where they have more double tides than in the UK. See the references to the criteria of Hicks in section 5.3 of Woodworth (2017).

p13, 4 - spring-neap

p14, 3 - the title of this paper is capitalised when Nunes and Lennon for example is not. Also doi's should be added for as many as possible.

p15, line 3 of caption - time difference in hours

line 6 - add a comma ... to being formed,

More general comments:

p12, 29-31 - this sentence is not true and you must remove or reword it. There are far

more than 'just a few places' - we have mentioned already that there are many in the Channel for example. Also this sentence devalues your own reasons for writing this paper! There are important practical reasons for understanding double highs and lows, for example in the need to understand the difference between reported MTL and MSL measurements, as discussed in Woodworth (2017).

Also, I don't agree with the justification that you do give for doing the work - the classification of tides. First, the definition of form factor that you give on p12-13 is the wrong way up - the form factor is usually defined as (K1+O1)/(M2+S2). (There is a typo in the equation bottom-right on p77 of Pugh and Woodworth, 2014 where it is also shown the wrong way). But, second, form factors like this are never going to be complicated by people further to take into account the temporal dependence such as you suggest - you may as well just look at the actual data. Form factors are just handy, simple descriptors of complicated situations, like climate indices.

Personally, I would rewrite the whole Conclusions section to review what you have actually done in the paper regarding the Doodson criterion and leave it at that. If you must mention the tides classification business then don't go into detail. Also mention how the maths changes with regard to double lows - I guess phi just moves to phi+180 - but double lows are of just as much interest. Also you could mention that, while you have extended the original Doodson criterion, and extended the discussion to when b/a is large, you have not covered all situtations - for example, on page 5-6 you just gave up when phi gets larger than an hour.

p10, section 3.2 and p12, about line 24 - this underlines the limitations of using real data in a paper like this. This paper is basically a game of playing with combinations of cosines, and the discussion would be much cleaner if you had used simulated data e.g. tidal predictions for Port Ellen based on harmonic constants derived from observations and readily available from BODC. I guess for scientists it is always attractive to use real data, and you could do that as well.
Anyhow, I enjoyed reading it.

Phil Woodworth

Extra references mentioned above:

Pugh, D.T. and Woodworth, P.L. 2014. Sea-level science: Understanding tides, surges, tsunamis and mean sea-level changes. Cambridge: Cambridge University Press. ISBN 9781107028197. 408pp.

Woodworth, P.L. 2017. Differences between Mean Tide Level and Mean Sea Level. Journal of Geodesy, 91, 69-90, doi:10.1007/s00190-016-0938-1.

---

## Author Comment (AC1) · 2 May 2017

A reply to all referees and and updated version of the manuscript are attached as supplementary information.

Please also note the supplement to this comment:
http://www.ocean-sci-discuss.net/os-2017-12/os-2017-12-AC1-supplement.zip

---

## Author Comment (AC2) · 2 May 2017

A reply and updated version fo the mansucript can be found in the supplementary information.

Please also note the supplement to this comment:
http://www.ocean-sci-discuss.net/os-2017-12/os-2017-12-AC2-supplement.zip

---

## Author Comment (AC3) · 2 May 2017

A reply and updated mansucript can be found in the supplementary information.

Please also note the supplement to this comment:
http://www.ocean-sci-discuss.net/os-2017-12/os-2017-12-AC3-supplement.zip

—————————————————

---

## Author Response (AR1)

**General reply**

We thank all three reviewers for their useful comments. Our replies to all reviewers are included below, and a revised version of the manuscript, with the changes implemented, have been uploaded.

It has been noted that OS asks us to include the reviewers' comments, our reply, and then an annotated version of the manuscript into one file. Below, we have included out replies immediately after each reviewer's comment. Also, we are using LaTex to produce our manuscripts and have had issues getting latexdiff to work. Instead, we have included two separate annotated manuscript versions. The first, in red, is the original submission, with red markup being deleted text. The second manuscript is the revised version with added text in green. We apologies for our inapt handling of latexdiff and hope our solution is acceptable to you.

It is our hope the paper is now suitable for publication in OS.

Dr J.M. Green

Corresponding author.

**Anonymous Referee #1**

General

This is a well written and appropriately brief account of how tidal harmonic constituents may add to give double high or low waters. The authors extend the simple Doodson criteria to cases of non-optimal phase relationships, and illustrate the theory with an example from Port Ellen in the Scottish Inner Hebrides.

Summing harmonics is essentially an exercise in trigonometry, rather than an investigation of tidal Kelvin wave dynamics. A regional study of wave generation and propagation around the amphidrome would be potentially more enlightening. Double high waters occur in a very few places worldwide, almost always associated with tidal amphidromes, where the range is near to zero. As such, as the authors state, this is "not a big topic" in tidal studies. Nevertheless, their extension of Doodson's simple formula does take it a stage further, and also shows how difficult and unrewarding ever further elaboration of the criteria through the trigonometry approach becomes; the paper should certainly be published for this reason, as well as for the intrinsic analysis.

Specific comments.

Line 1 Doodson's was a minimum criterion.

**Amended**.

L12 here "neap" is used for the time of minimum local tidal ranges, though because of amphidrome movements during the spring-neap cycle, this may not be near first and last lunar quadrature when small neap ranges occur in the tidal forcing.

**That is true, and it is quite likely that the minimum in range at Port Ellen does not occur at quadrature. We will stick with our wording though, but have also added a comment explaining what we mean.**

L25 "correct" is not the right word. Maybe "an appropriate" is better.

**Amended.**

L1 a third reason for higher harmonics is streamline curvature around indented coasts and sansbanks.. Godin is a very obscure reference for such a universal truth.

**Text added and reference updated.**

L10 replace "stringent" with "minimum"

**Amended.**

L 13 explain why phi has a negative sign

**So that D2 has a maximum for t=0 and the Dn harmonic a minimum at t=phi. Clarified in the text.**

Bottom line be consistent in Eq. or eq. there may be other examples.

**Amended.**

Caption last line is incorrect grammar ..should be " when neither a stand, nor a double. . ."

**Amended.**

L 6 1/9 no1 1/32

**Amended.**

L5 an enlarged plot of selected days would help. Fig 1 is a bit too compressed.

**We think the reviewer might mean that fig 4 is a bit too compressed. However, we feel that little is added by including details of individual days.**

L24. Say what is the sampling interval (15 minutes) and correct the source specification. It is not PSMSL. PSMSL is an international body located at NOCL which publishes monthly and annual mean sea levels for IAPSO. UK 15-minute sea level data comes from BODC (also at NOC Liverpool) under contract from DEFRA, SEPA etc. There is a specific form of acknowledgement required. . .see their web site. As DEFRA through the Environment Agency, and SEPA pay a lot of money for this measurement programme they like to get some credit.

**We have updated the source specification – thank you for pointing out this glaring mistake. We have not, however, found a specific acknowledgment, but updated ours to point to the correct page.**

L25 show the amphidrome in Figure 4

**Done, although we assume the reviewer refers to Fig 3?**

L27 this is similar to Courtown tides and worth a brief comparison (see Pugh 1982)

**Added.**

L28 180 degrees not 1800.

**Amended.**

L10 here the authors seem to be calling "Harmonic analysis" narrowly the fitting of daily curves. D and W actually used Y1, Y2..etc. Say so.

**We would argue that the fitting, in a least-squares sense, of the fundamental period and its harmonics to 25 hours of data is indeed a harmonic analysis, and we used the same method as Airy and D&W,**

L3 If they are 15-minute readings, a better symmetry would have been to go from 2330 in day -1 to 0030 on day +1.

**Possibly, but the results would have been the same.**

L7 "treated by a full harmonic tidal analysis.."

**Amended.**

L11 A plot of each of the daily D1, D2, D4 and D6 amplitudes and a discussion would be useful here to understand what is happening.

**We have added such a figure and are referring to it in the text.**

L14 Note that observations include met effects, seiches and instrument noise.

**Added.**

The line darkening doesn't work well. Surely web publication allows full colour?

**Amended.**

L8 HA seems now to be called Fourier Analysis. Non-mathematical readers may find this alternative naming confusing. Be consistent.

**Amended.**

L15 specifically the advantage for D6 is (9/4)*(44/26). . .say so.

**Excellent suggestion – now included.**

L 32 and noise from Met, seiching etc..

**Added.**

L18 This is an important discussion and it would be useful to plot the D4 and D6 amplitudes against D2 here. See Pugh and Woodworth 2014 for what happens at Southampton. This interdependence of the basic and higher harmonics has big influences on the turning points occurrences.

**The new figure, figure 6, confirms that D6 depends on D2 but D4 does not. We have opted to not include a scatter plot of this to keep the paper length down, but a statement has been added to the text.**

L21 not sure why phase is said to be of "most importance". . .it's both amplitudes and phases. . .

**Clarified**

"perhaps in general" is just too vague and unjustified.

**Amended.**

L8 Doodson and Warburg discuss the "three plus harmonics" case, on page 222, for amplitudes only.

**Added.**

L21 this just confirms how unrewarding further complicating the criteria would be.

**Indeed.**

L1 and 2 Replace the two "and"s in parenthesis with + . It's in the definition.

**Amended.**

The suggestion that the designation semidiurnal, mixed and diurnal needs redefining to allow for daily changes is valid, but a needless complication. The whole point about this basic descriptor is its simplicity. As an example of how a time dependent Form Factor might go, at lunar zero declination the D1 is near zero, and so the regime form factor ratio could be zero globally (ok, ignoring solar declination), but how would that improve our simple descriptor of local tides at a port?

**The form factor is simple but it doesn't work because the relative amplitudes change with time. A revised form factor allowing for this WOULD be useful, however, and not greatly more complicated (in the same way that the revised Doodson criterion reveals things not shown by the original version). We have opted to keep the text as it stands.**

In Table 1 the values of $R^{**2}$ have negative values sometimes. Is it worth considering the implied virtual values of r? might be worth a comment on page 3.

**This is an interesting thought, but we feel there there is very little information in interpreting the negative $r^{**2}$. In short, we have not found any useful information in the negative values of r2 beyond the fact that double tides are precluded. We have added a comment.**

**Anonymous Referee #2**

When I began reading this paper, I was inclined to send in a favourable review. However, when I tried checking the mathematics, I ran into so many careless errors that I finally gave up. Please have the authors revise their manuscript and check it carefully this time. After that, I will then review the paper.

Specifically, they need to address the following:
Equation (5) does not follow from Eq (4).

**It does; the typo confusing the reviewer has been amended (see next comment), and a clarification added.**

The line after Eq (5) refers to a term in brackets. What brackets?

**Typo – now included (there was also a factor phi missing).**

Eq (6) is clearly wrong.

**Indeed it is – amended.**

Page 5, line 4, a new variable "f" is introduced but not defined.

**Typo, should be \phi – thanks for spotting it!**

Figure 1 has a floating label "D" but it doesn't point to anything.

**Fixed.**

In their revision, the authors may also wish to consider a previous paper by Godin, published in 1993 in the Deut Hydrogr Zeitschrift.

**Thanks for pointing this out – it has been added.**

**Referee #3 (Professor Woodworth)**

This is an entertaining little paper in the style of papers by tidal scientists a century ago. It discusses the 'Doodson criterion' whereby harmonics (M4, M6 etc.) of the main tidal constituent (M2) can generate a double high or low water such as occurs at Southampton. The criterion, which is mentioned in Doodson and Warburg (1941, p.221) and more recently in Pugh and Woodworth (2014, p.138), is clearly stated to reflect the ideal situation of producing a double tide, when the phase relationships are optimal. The present paper aims to extend consideration of that special situation into the more general case when there are different relative amplitudes and phases between the different constituents.

I found it to be quite an educational exercise and it overlaps with some of my own work as will be apparent from the comments below. So I have no objection to its publication. I have some small comments and a couple of more general ones.

Small comments:

p1, first para of the Introduction -I found this wording disappointing as the authors have just paraphrased p133 of Pugh and Woodworth (2014). I think if I had been them I would have worded it to say that, given the spatial scales of the harmonics are shorter than that of M2 (as they discuss below), then whenever you have a place like S'ton with a double high then there is probably a place with a double low a short distance away (Weymouth in this case). For example, if you travel along the French/Belgian/Dutch coast, double highs are interleaved with double lows several times, not just at isolated places like Den Helder. This becomes most evident using tide models of course.

Then some mention of other places such as the US would be appropriate by all means. There is discussion of Weymouth and the Doodson criterion in Woodworth (2017). The Redfield reference for New England corresponds to what Woodworth (2017) calls the 'Newport enclave'.

**It was never our intention to paraphrase, and we do apologise if it came across that way. The introduction has been rewritten and the suggested locations added. Also, thank you for pointing out Woodworth (2017).**

Finally, I suspect there may be a location not too far from Port Ellen where there are double lows at times. Maybe you could look into that.

**It is not immediately evident where this would be, and we leave it for a future study. A comment has been added in the discussion.**

p2, 14 - value of it

**Amended**

same line - spring-neap

**Amended**

19 - I wouuld say 'about 4 hours'

**Amended**

25 (Nunes and Lennon, 1986)

**Amended**

26 - a should be in italics

**Amended**

28 - This wording might confuse some people. It is quite reasonable for you to investigate when the harmonics are larger than M2, and it is an extension to Doodson's work, but it is not an extension to the Doodson criterion as such, which was explictly to do with harmonics smaller than M2.

**Clarified.**

p3, 7 - $b/a = 0.25$ and $a = 1$ metre

**Amended**

21 - you should state that tprime is the time of the dip

**Done now.**

23 - I think this should read 'in the (semidiurnal) high water'

**Changed.**

eq.6 - b/b should be b/a

**Amended**

p4, fig.1a - the D seems to at 14 hours and not at the time of the dip at about 12.5 hours. Can you have an arrow pointing to the dip?

**The arrow we mention has been added – it had been removed by the figure gremlins.**

In the caption, mention that time is measured from the peak of the M2 tide, as you do in the text.

**Done.**

p5, 4 - f should be phi

**Amended**

6 - 1/32 should be 1/9

**Amended**

7-8 - 'at least about' sounds like you don't know what it really is. drop 'about'.

**Done.**

I had to read this paragraph twice as it was not clear at the start that what was being discussed was the approximate parameterisation - it became clear at line 9, and it is clear in fig.2 caption. Perhaps review the wording here.

**Clarified**

p6, 3 - I wondered if a subheading was appropriate here as the discussion now switches to the opposite situation of large b/a.

**Possibly, but we prefer to not add another level of sections.**

section 3 - I don't know where you got the data from - it may well have been from someone in PSMSL, but PSMSL is not usually associated with the UK 15-minute data. I would credit the source as BODC (www.bodc.ac.uk) or NTSLF (www.ntslf.org), or maybe you got it from the IOC Facility (www.ioc-sealevelmonitoring.org).

**This is a mistake on our behalf and it has been amended in response to Ref 1. It is now credited to NTSLF.**

28 - 1800 should be 180 deg

**Amended.**

drop the last sentence this page - this repeats from above.

**We have actually retained it, and instead expanded on it by adding Courtown as well.**

p7, Fig.3 - I would drop this. The exact location of Port Ellen is of no importance to the reader, all he needs to know is that it is a place with similar M2 and S2 amplitudes. If you want to keep it, I would make it more useful by adding the M2 tidal map from a model - I am sure Dr. Green has one.

**We disagree, and have kept the figure, with one compromise: we have added the location of the amphidromic point (as also suggested by Ref1).**

6 - tides should be times

**Amended.**

Fig5 caption - Observed (bold black) and fitted (grey) ...

**Fixed – the figure is now in colour and the caption updated.**

line 3 - there is no Fig.3a. You mean 4 or 5a?

**Should be 4, updated.**

p11, 14-17 from 'We expect' to 'critical lines'. Why do you expect this etc? It is not obvious to me.

**The revised lower limit of the Doodson criterion is $(b/a)r^2>1$, where $r^2$ is a new parameter. Figure 2 shows a graphical solution to that inequality (with curves for an analytical solution and an exact numerical one). If we are above these curves, then the inequality is satisfied and we expect a double high water to form (incidentally, the same is true for double low waters, phi becomes the time difference between D2 low and Dn high). We've coded the tidal cycles at Port Ellen (black, grey, white) and they generally agree with the curves. We feel this is in the text, but have clarified it further.**

24-31 - make it clear that these limits derived from table 1 apply to an M2/M6 situation and not an M2/M4 situation.

**We disagree. The general case for a DHW we are proposing is $n^2> (b/a)r^2>1$ and this applies to ANY harmonic as long as $r^2$ is calculated appropriately. In the case of Port Ellen, $r^2$ has been calculated for D2 and D6 tides as we think these are the ones responsible for the DHW as explained in the text. The theory holds for any combination, however.**

p12, 18-19 - there are already criteria for defining double tides in some operational agencies e.g. in the US (NOAA), where they have more double tides than in the UK. See the references to the criteria of Hicks in section 5.3 of Woodworth (2017).

**Thank you for pointing this out – this sentence has been deleted.**

p13, 4 - spring-neap

**Amended.**

p14, 3 - the title of this paper is capitalised when Nunes and Lennon for example is not.

Also doi's should be added for as many as possible.

**Fixed.**

p15, line 3 of caption - time difference in hours

**Amended.**

line 6 - add a comma ... to being formed,

**Amended.**

More general comments:

p12, 29-31 - this sentence is not true and you must remove or reword it. There are far more than 'just a few places' - we have mentioned already that there are many in the Channel for example. Also this sentence devalues your own reasons for writing this paper! There are important practical reasons for understanding double highs and lows, for example in the need to understand the difference between reported MTL and MSL measurements, as discussed in Woodworth (2017).

**Fair comment, the text has been updated accordingly.**

Also, I don't agree with the justification that you do give for doing the work - the classification of tides. First, the definition of form factor that you give on p12-13 is the wrong way up - the form factor is usually defined as (K1+O1)/(M2+S2). (There is a typo in the equation bottom-right on p77 of Pugh and Woodworth, 2014 where it is also shown the wrong way). But, second, form factors like this are never going to be complicated by people further to take into account the temporal dependence such as you suggest – you may as well just look at the actual data. Form factors are just handy, simple descriptors of complicated situations, like climate indices. Personally, I would rewrite the whole Conclusions section to review what you have actually done in the paper regarding the Doodson criterion and leave it at that. If you must mention the tides classification business then don't go into detail.

**We have amended the text and followed the recommendation of the reviewer.**

Also mention how the maths changes with regard to double lows - I guess phi just moves to phi+180 - but double lows are of just as much interest. Also you could mention that, while you have extended the original Doodson criterion, and extended the discussion to when b/a is large, you have not covered all situations - for example, on page 5-6 you just gave up when phi gets larger than an hour.

**This is now included.**

p10, section 3.2 and p12, about line 24 - this underlines the limitations of using real data in a paper like this. This paper is basically a game of playing with combinations of cosines, and the discussion would be much cleaner if you had used simulated data e.g. tidal predictions for Port Ellen based on harmonic constants derived from observations and readily available from BODC. I guess for scientists it is always attractive to use real data, and you could do that as well.

**True, but we feel that the paper has a significant message, and in many other locations the only data available are direct measurements of sea level, perhaps for short periods of time. So, we have opted to not bring in the BODC cefficients in this paper.**

Anyhow, I enjoyed reading it.

Phil Woodworth

Extra references mentioned above:

[revised manuscript text omitted]